# The Effect of In Vitro Electrolytic Cleaning on Biofilm-Contaminated Implant Surfaces

**DOI:** 10.3390/jcm8091397

**Published:** 2019-09-06

**Authors:** Christoph Ratka, Paul Weigl, Dirk Henrich, Felix Koch, Markus Schlee, Holger Zipprich

**Affiliations:** 1Department of Prosthodontics, Goethe University, 60590 Frankfurt am Main, Germany; ratka@med.uni-frankfurt.de (C.R.); weigl@em.uni-frankfurt.de (P.W.); 2Department of Trauma, Hand & Reconstructive Surgery, Goethe University, 60590 Frankfurt am Main, Germany; d.henrich@trauma.uni-frankfurt.de; 3Private Practice, Department of Maxillofacial Surgery, Goethe University, 60590 Frankfurt am Main, Germany; felixpkoch@gmx.de (F.K.); markus.schlee@32schoenezaehne.de (M.S.)

**Keywords:** dental implant, biofilm, infection, perio-prosthetic joint infection, periimplantitis, electrolytic cleaning

## Abstract

Purpose: Bacterial biofilms are a major problem in the treatment of infected dental and orthopedic implants. The purpose of this study is to investigate the cleaning effect of an electrolytic approach (EC) compared to a powder-spray system (PSS) on titanium surfaces. Materials and Methods: The tested implants (different surfaces and alloys) were collated into six groups and treated ether with EC or PSS. After a mature biofilm was established, the implants were treated, immersed in a nutritional solution, and streaked on Columbia agar. Colony-forming units (CFUs) were counted after breeding and testing (EC), and control (PSS) groups were compared using a paired sample *t*-test. Results: No bacterial growth was observed in the EC groups. After thinning to 1:1,000,000, 258.1 ± 19.9 (group 2), 264.4 ± 36.5 (group 4), and 245.3 ± 40.7 (group 6) CFUs could be counted in the PSS groups. The difference between the electrolytic approach (test groups 1, 3, and 5) and PSS (control groups 2, 4, and 6) was statistically extremely significant (*p*-value < 2.2 × 10^−16^). Conclusion: Only EC inactivated the bacterial biofilm, and PSS left reproducible bacteria behind. Within the limits of this in vitro test, clinical relevance could be demonstrated.

## 1. Introduction

Growing numbers of inserted dental implants correlate to an increasing number of infected implants [1]. Periimplantitis (PI) is defined as an inflammatory process affecting both periimplant soft and hard tissue. Craterlike bone defects, ongoing bone loss, pus, and bleeding on probing (BoP) are clinical parameters that have to be present to justify the diagnosis of PI [2,3]. PI correlates to bacterial biofilms growing on implants or abutments [4]. In view of the various definitions of PI and the lack of agreement in the dental community about an acceptable threshold of bone loss, there is no consensus on when pathology starts and how PI can be diagnosed precisely. Hence, there is no consensus about prevalence data [5,6,7]. A proper treatment modality is lacking [8]. Whether the bacterial biofilm is the single causal factor or only correlates to PI [9] is still a matter of discussion. This debate about etiology is not merely an academic question, but may influence the success rate of therapy, because of possible differences in specimen susceptibility and uncorrectable surgical or mechanical obstacles. In any event, biofilm needs to be removed to prevent progression of disease, or to treat PI successfully. As implant surfaces exposed to the oral cavity are immediately colonized by bacteria, the surfaces need to be re-osseointegrated for good long-term results [10]. 

Infection of plates or other orthopedic devices, or endoprostheses caused by polymicrobial biofilms, is a major challenge. The average infection rate of implants in the shoulder, knee, and hip is less than 2% in most treatment centers [11,12]. A recent meta-analysis demonstrates an overall cumulative incidence of endoprosthetic joint infection across all studies of 0.78% [13]. It has been reported that, in the United States, more than 1 million implants placed in shoulders, knees, and hips need to be replaced every year due to infections [14]. A systematic review showed a reinfection rate of 7.6% (95% CI 3.4–13.1) of hip and knee replacements [15]. Polymicrobial biofilm on orthopedic implants shows high resistance to immune defense, and will result in local inhibition of granulocyte activity around the implant. This effect is referred to as frustrated phagocytosis, which leads to degranulation, reduced ingestion ability, and production of oxygen radicals in the granulocytes [16]. The standard treatment protocol for infected orthopedic implants is replacement in a one- or two-stage procedure, with a success rate of 80−90% [17,18]. Even early infections within the first 30 days after implantation that are treated by aggressive debridement and long-time antibiotics yield success rates of only 71% [19,20]. 

In most cases, all treatment modalities fail to cure infected implants, requiring replacement of the implant or causing at least an inadequate failure rate and immense costs. Therefore, it is necessary to search for another approach to decontaminate infected implants.

Several ablative methods to remove biofilms, such as mechanical debridement with curettes, brushes, lasers, cotton pellets, cold plasma, or air-powder sprays, combined or not combined with disinfecting or antibiotic agents, have been discussed. Re-osseointegration of formerly infected dental implant surfaces was reported in animal studies for between 39% and 46% of the surface [21]. There is no evidence of whether and to what percentage re-osseointegration occurs clinically. A recent review of the literature demonstrated the equality of all published methods, and no method seemed able to achieve stable results over time [22]. Follow-up revealed recurrence of the disease in up to 100% for some methods after one year [8]. A powder-spray system (PSS) is commonly used to clean implant surfaces. Erythritol (size 14 µm) or glycine (size 25 µm) particles are accelerated by air pressure (2 bar). When the particles impact at an angle of 30–60° at an ideal working distance of 3−5 mm, the kinetic energy of the powder removes biofilm, according to the manufacturer’s manual. Several animal studies investigating re-osseointegration after cleaning with a PSS proved incomplete re-osseointegration [21]. Improvement of periimplant parameters, such as BoP, probing depth, and pus was clinically proven after the PSS, but re-osseointegration was not proven. Furthermore, the PSS failed to demonstrate superiority over any other treatment modality [23,24,25]. Possible reasons for this incomplete cleaning efficacy might be craterlike bone defects, with limited access and thus an improper working angle and distance of the device, the macro- and micro-design of implant surface, and oversized particles for much smaller bacteria hidden in the microstructure of textured implants. In an in vitro test, implants contaminated with *E. coli* were treated with a continuous current of 0−10 mA. Two implants were loaded as either cathode or anode, resulting in reduced numbers of bacteria on both implants. On anodic implants, a current of more than 75 mA caused complete killing of the bacteria [26]. Zipprich et al. loaded the implant as a cathode, and a platinum-coated titanium acted as the anode [27]. In this in vitro trial, the implants were exposed to a sodium-iodine solution buffered by lactic acid to avoid alkalization of the salt solution [27]. Hydrolytic splitting of water produces hydrogen cations as long as the implants are loaded. These hydrogen cations penetrate the biofilm, and after an electron has been taken from the implant surface, hydrogen forms between the implant surface and the biofilm. The hydrogen bubbles remove the biofilm. While all the other discussed methods for removing a biofilm are ablative, the electrolytic approach presented here works under the biofilm, directly on the implant surface, regardless of surface characteristics or alloy. Our results support the finding that the potassium iodine solution allows a higher current than the previously published sodium chloride solution [27]. This results in higher hydrogen production and a possible direct bactericidal effect. A cooperating working group examined the exact working mode of this method and proved its effectiveness in killing bacteria [28]. The current gold standard in dentistry is to remove biofilm with a PSS. These positive preliminary results for an electrolytic approach (EC) encouraged the authors to compare the cleaning effect of air-powder-water spray as an accepted method of cleaning contaminated implant surfaces with the new electrolytic approach in vitro.

## 2. Materials and Methods

### 2.1. Null Hypothesis

The null hypothesis is that the electrolytic approach works equally as well as PSS cleaning. 

### 2.2. Preparation of Test Implant and Grouping

For the study, test implants were designed and produced by the authors to be as close as possible to clinical reality in implant and orthopedic surgery. The test specimens were made of grade 4 and 5 titanium, and consisted of a test and a carrier part (Figure 1). The design of the test part was similar to a standard, parallel-threaded dental implant (∅ 4.0 mm/L = 11.0 mm pitch = 0.6 mm) with different surface modifications. The carrier part was machined to serve as the electricity conductor and to fix the test specimen in the experimental set-up without influencing the area of interest of the test area. Furthermore, the test and carrier part could be separated at a predetermined breaking point to avoid bacterial contamination of the test area during handling.

Six groups of implants (*n* = 12 per group, 72 in total) were investigated. Ten per group were treated (60 in total), while two served as negative controls and were not treated (12 in total):

Group 1: titanium grade 4 + sandblasting and acid-etching + electrolytic cleaning;

Group 2: titanium grade 4 + sandblasting and acid-etching + air-powder-water spray cleaning;

Group 3: titanium grade 5 + sandblasting and acid-etching + electrolytic cleaning;

Group 4: titanium grade 5 + sandblasting and acid-etching + air-powder-water spray cleaning;

Group 5: titanium grade 4 + anodic oxidation + electrolytic cleaning;

Group 6: titanium grade 4 + anodic oxidation + air-powder-water spray cleaning. 

Groups 1, 3, and 5 (test group) were treated with an electrolytic treatment, and groups 2, 4, and 6 (control group) were treated with a PSS.

### 2.3. Saliva Collection

In order to ensure a diversity of microorganisms, saliva was collected from three volunteers (different ages, 2 male, 1 female, healthy, no medication or drugs). The collected saliva (3 mL) was pooled, centrifuged to spin down any large debris and cells (2600 *g* for 10 min), and the liquid supernatant was mixed with 0.5 L nutrition solution culture (Bacto Tryptic Soy Broth, Becton Dickenson, Heidelberg, Germany), and incubated under 37 °C on the rocking device for 48 h, as previously described [29]. The solution was then filled into tubes (1.5 mL, Eppendorf Safe-Lock Tubes, Eppendorf, Hamburg, Germany) and frozen at −18 °C to provide identical samples for all further tests.

### 2.4. Establishing the Biofilm

To breed the biofilm on the implants, the required bacterial samples were thawed and bred with 300 mL nutrition solution in a beaker at 37 °C for 24 h. The carrier parts of the sterilized implants were masked to reduce bacterial contamination and fixed on a base plate. The experimental set-up was then immersed in the beaker with the culture medium to initiate formation of the biofilm. The beaker was placed in the rocking incubator for 14 days at a temperature of 37 °C. Previous tests with daily checks with a scanning electron micrograph (SEM) had demonstrated that 14 days were necessary to achieve a mature, multilayer biofilm (Figure 2). Every two days, the bacterial culture medium was replaced with freshly prepared nutritional solution. 

### 2.5. Electrolytic Cleaning (Test Group)

The implants in groups 1, 3, and 5 were cleaned by an electrolytic process. Figure 3 shows the construction of the electrolytic chamber. After demasking of the biofilm-coated test implant with a customized ejector, the implant was mounted on the bottom of the electrolytic chamber with a customized connector. The test implant served as the cathode. The connector had an inner titanium sleeve (titanium grade 4) covered by an insulating polyetheretherketone (PEEK) carrier. The inner titanium sleeve was connected to the electric source, and the PEEK carrier masked the carrier part of the test implant for only three quarters. This design was chosen to guarantee that it was possible to separate the test specimen using a customized tool, without touching the PEEK carrier, which was not been cleaned during the electrolytic procedure. The connector was mounted in a polytetrafluoroethylene (PTFE) socket to ensure stability of the electrolytic chamber. 

After the cathode was fixed, the glass cylinder of the chamber was mounted, and 35 mL of electrolyte (sodium iodide (200 g/L), potassium iodide (200 g/L), L(+)-lactic acid (20 g/L), and water (800 g/L)) was poured into the chamber. Fresh electrolyte was used for each test. 

The anode (titanium grade 4) was coated with platinum to prevent passivation of its surface. The anode was fixed in a hollow PTFE tube to ensure direct contact of the platinum-plated part with the electrolyte (∅ 5.0 mm/L = 4.0 mm). After the other end of the PTFE tube had been mounted in the cap of the chamber, the anode and cathode were connected to a voltage source (PeakTech 6060, PeakTech, Ahrensburg, Germany). 

A voltage of 6 V was applied for 5 minutes, resulting in a current of up to 1100 mA. 

After the cleaning process was finished, the cap of the chamber was removed, and the used electrolyte was poured into a beaker. The biofilm floated as a visible layer on the fluid in the beaker, and was pipetted with some fluid to prove the vitality of the collected bacteria (waste from electrolytic cleaning).

### 2.6. Air-Powder-Water Spray (PSS) Cleaning (Control Group)

The implants in groups 2, 4, and 6 were treated by a PSS (BA 8000, Mectron GmbH, Cologne, Germany). To ensure a reproducible and comparable cleaning effect, a test apparatus was assembled, as shown in Figure 4. The aim was to reproduce the working angle and distance according to the manufacturer´s operating manual (45° and 10 mm). The test implants were mounted on a sterile PEEK carrier (Figure 4, in orange), which masked three quarters of the carrier part and was then fixed in a PTFE connecting bar (Figure 4). The holder provided the same working angle and distance for all the tested implants. The holder and its fixed handpiece were moved manually along the implant axis while the test implant rotated, being driven by a motor around its axis (40 rpm) to ensure proper and repeatable cleaning of the implants (Figure 4). Sterile water (Aqua ad injectabilia, Braun, Melsungen, Germany) and aminoacetic acid powder (glycine powder, Mectron GmbH, Cologne, Germany) were used with an air pressure of 2 bar being applied for 60 s. The waste liquid produced during the cleaning process was collected in a beaker to prove the vitality of the removed biofilm (waste from air-powder-water cleaning). 

### 2.7. Rinsing and Test Sample Incubation

After cleaning, the test and control implants were rinsed with air-water spray for 5 seconds (distance of 5 cm) and dried with oil-free air. The test parts were then broken off at the predetermined breaking point in a sterile manner, transferred to Eppendorf tubes filled with 1.5 mL nutritional solution, and incubated at 37 °C for 24 h. Ehrensberger et al. demonstrated that no additional colonies grew when incubation time was extended 18 h. [30] 

### 2.8. Analysis of the Bacterial Growth

The solution in each Eppendorf tube was pipetted, transferred and streaked with an L-spatula on blood agar (Columbia Blood Agar-VWR, BDH Chemicals, Leuven, Belgium). The plate was incubated at 37 °C for 24 h. 

The rest of the contents of the Eppendorf tube was gradually diluted in six steps to 1:1,000,000. Each step was also spread on a blood agar plate and treated as described above. In total, seven dilution steps were bred (210 in test group, 210 in control group, 420 in total).

The change in the plate condition was photographed after 24 h. The number of colony-forming units (CFUs) were counted using the software ImageJ (version 1.51u, National Institutes of Health, Bethesda, MD, USA). The difference between the two Poisson rates has been applied to evaluate the difference in bacterial growth.

### 2.9. Sterility of the Experimental Set-Up (Positive Control)

To prove the sterility of the different experimental set-ups, the air-powder-water spray was directed to blood agar plates (two per group, six in total). The sterility of the nutritional solution was proved by breeding it at 37 °C for 24 h, and then spreading it on a blood agar plate. The agar plates were incubated at 37 °C for 24 h, and then CFUs were counted.

### 2.10. Quality of Biofilm (Negative Control)

To prove the quality of the biofilm, two test implants from each group (12 in total) served as the negative control and were not cleaned. After biofilm formation, the nutritional solution was rinsed away with sterile water. One implant per group (six in total) was rinsed with sterile water (Aqua ad injectabilia, Braun, Melsungen, Germany) and dried in air for 24 h. The usual fixation in ethanol was not done, to avoid washing-out of the matrix. The samples were gold-coated and examined using a Philips SE XL30 (LaB6 cathode) scanning electron micrograph (SEM) with 20 kV power and a spot size 4 to check biofilm formation and quality. One implant per group (six in total) was incubated in an Eppendorf tube with 1.5 mL nutritional solution and bred at 37 °C for 24 h. In anticipation of floating bacteria, the nutritional solution was spread on blood agar and bred at 37 °C for 24 h. Colony forming units (CFUs) were then counted (Figure 5). 

### 2.11. Vitality of Bacteria after Cleaning

To prove the vitality, the waste from both cleaning methods (electrolytic and PSS) was spread on blood agar and bred at 37 °C for 24 h. Then CFUs were counted. 

### 2.12. Counting Colonies

The number of colonies that had grown on the blood agar plates were manually counted using the ImageJ (version 1.51u) software.

### 2.13. Statistics

The number of colonies that had grown on the blood agar plates after a dilution to 1:1,000,000 where manually counted and processed by the ImageJ (version 1.51u) software.

Continuous variables are reported as mean ± standard deviation. All groups were tested for normality by the Shapiro-Wilk test. Comparisons between the two cleaning methods were performed with the paired sample *t*-test. Comparisons between the surface and material groups for PSS were performed with the Kruskal-Wallis test. 

A two-sided *p*-value of ≤0.05 was considered statistically significant. Statistical analysis was performed with R (R Foundation for Statistical Computing, Vienna, Austria).

## 3. Results

### 3.1. Quality of Biofilm (Negative Control)

The formation of a mature layer of biofilm completely covering the surface of all six tested implants was demonstrated upon checking by scanning electron micrograph (SEM). Six implants were not treated, but were incubated as described. CFUs could not be counted because the blood agar plate was totally covered with a closed layer of bacteria. 

### 3.2. Colony-Forming Unit Count

In groups 1, 3, and 5 (test groups, electrolytic cleaning), 210 blood agar plates were evaluated. Not a single CFU could be counted on any blood agar plate. Neither dilution (Figure 6) nor alloy nor surface had any influence on this observation. 

The cultivation of the waste liquid collected from the electrolytic cleaning process also showed no bacterial growth.

In the control groups (2, 4, and 6; PSS; 210 agar plates), the growth of bacteria was observed on every agar plate with any dilution grade (Figure 7), irrespective of dilution, alloy, or surface. CFU counting was not possible in the first five steps of dilution (from 1:1 to 1:100,000), because the bacteria grew over the whole agar plate. CFU counting was possible at a dilution of 1:1,000,000. In all PSS groups, more than 200 CFUs were documented (group 2 = 258.1 ± 19.9, group 4 = 264.4 ± 36.5, group 6 = 245.3 ± 40.7) (Table 1). The cultivation of the collected waste liquid always showed bacterial growth.

### 3.3. Sterility of Experimental Set-Up (Positive Control)

No sign of bacterial growth was observed on the blood agar after PSS was sprayed directly onto blood agar plates. No sign of bacterial growth was observed on the blood agar after the nutritional solution was bred. This indicates that the air-powder-water spray system and the prepared nutritional solution were free from bacterial contamination. 

## 4. Discussion

The application of voltage to reduce bacterial load has been described several times in the literature. Ehrensberger et al. [30] evaluated the influence of cathodic voltage-controlled electrical stimulation (CVCES) to eradicate methicillin-resistant *S. aureus* (MRSA) from titanium surfaces in vitro and in Long-Evans rats. The application of −1.8 V for 1 h reduced MRSA CFU counts by 87% in bone and 88% in titanium, compared to the open-circuit potential in vitro, and by 97% in titanium in vivo. No histological changes were detected in the rat model. CVCES increased the interfacial capacitance (from 18.93 to 98.25 µF/cm^2^) and decreased the polarization resistance (from 868.25 to 108 Ω/cm^2^). Because of the negative charge of the titanium surface and the negative surface charge of *S. aureus*, attachment of the biofilm may be influenced. Furthermore, it has been discussed that the electric field on polarized electrodes generates anions by electrolytic splitting, which disrupts and pushes away the biofilm [24,31,32]. The major reaction at the cathode is the reduction of water or the reduction of oxygen:

Reaction 1: 2H_2_O + 2e^− →^ H_2_ + OH^−^

Reaction 2: O_2_ + 4^−^ +2H_2_O ^→^ 4 OH^−^

Blenkinsopp et al. proposed an altered transport of ions within the biofilm [33]. Portinga et al. demonstrated an extended attachment of bacteria to the biofilm after donation of their free electrons to the matrix. Free electrons on the cathode might disturb this adherence [34,31], To summarize, the antimicrobial effects seem to be related to voltage-dependent surface properties of the titanium.

Several authors [26,35,36] have reported that loading two contaminated dental implants (one as anode and the other as cathode) in NaCl as electrolyte significantly reduces the number of bacteria. The downside of the published method is that toxic chloric and hypochlorous acid are reaction products, and the pH in the periimplant area decreased or increased significantly. 

Zipprich et al. [27] introduced an electrolytic approach to remove biofilm from explanted implants by using a mixture of sodium iodide, potassium iodide, and water buffered by L(+)-lactic acid. Schneider et al. [28] investigated the underlying mechanism of this method of electrolytic cleaning. Several disinfecting agents, such as triiodide and hydrogen peroxide, were generated on the implant surface loaded as the cathode. The authors proved in an in vitro test that the major effect in removing the biofilm was the generation of hydrogen on the implant surface, which pushed away the biofilm mechanically. Based on these findings, the authors decided to prove the effectiveness and efficiency of this approach in this in vitro test. It is very difficult to perform a controlled in vitro study on a “real” biofilm in real periimplantitis cases, because its three-dimensional structure is destroyed while the implant is removed, and anaerobic bacteria would not survive this process. The nutritional and biological conditions of an in vivo biofilm will differ from biofilms used in this in vitro test. Such samples may harbor different mixtures of bacteria. This pitfall compromises all in vitro studies done on periimplant biofilms. On the other hand, it was demonstrated that microbiota associated with periimplantitis are quite variable and inconsistent [4].

Different methods of growing biofilms have been published. Most of the articles used single-species biofilms with very little relation to real periimplant biofilms. *S. aureus* is a colony-forming bacterium associated with infected dental or orthopedic implants, and was bred for several days on titanium surfaces [30,37]. Mohn used *E. coli*, which is not associated with periimplantitis [26]. Other authors harvested intraoral subgingival plaque and exposed titanium plates in a nutritional solution for seven days [38]. John et al. cultivated biofilms by exposing titanium plates fixed on splints for 48 hours in the oral cavity of two specimens [39]. In preparation for this study, we were not able to reproduce the claims of these authors to achieve a mature biofilm covering the complete implant surface in the published timeline, when using a model with intraoral bacteria. It took 14 days to achieve a multilayer mature biofilm covering the whole implant surface (Figure 2). 

To imitate micro- and macrostructure, we decided to use lifelike test implants. To carry such implants in the mouth for several days to accumulate bacteria would be unacceptable for humans. Consequently, we had to develop a realistic extraoral model to breed a biofilm. A single-species model was used, and did not form a complete monolayer of bacteria nor a multilayer biofilm. To achieve the most realistic diversity of microorganisms, the saliva was collected and pooled from three volunteers, bred extra-orally, and frozen to ensure the same biota for all the tests. In preparation for this study, we found out that it takes 14 days to achieve the expected quality and thickness of biofilm. In vitro, it could have been demonstrated that a cold plasma device can clean heling abutments more effectively than conventional cleaning [40].

No published method was able to remove all bacteria from implant surfaces in the clinical setting. Moreover, bacteria will re-colonize implant surfaces exposed to the oral cavity within a matter of hours [10]. The theory that elimination of the biofilm will cure periimplantitis is an attractive one [4]. Clinically, it may be necessary to remove the bacteria and extracellular matrix in a way that will allow re-osseointegration of formerly infected surfaces. 

We were able to demonstrate that no bacteria could be cultivated in the nutritional solution and powder used for a PSS, thus proving the sterility of the experimental set-up. The vitality of the bred biofilm was proved visually by SEM and by CFU counting after its cultivation on blood agar plates. As different titanium alloys and surface modifications are used in dental and orthopedic surgery, we tested titanium grades 4 and 5, as well as acid-etched, sandblasted, and anodized implant surfaces. Their effectiveness in the removal of bacteria was compared to a standard method (PSS). 

The results were unequivocally clear. It was not possible to breed bacteria after the implants had been cleaned by the electrolytic approach. Every implant cleaned by the air-powder-water spray contained so many bacteria that all the blood agar plates were totally overgrown with a lawn of bacteria. A single CFU could only be counted after dilution to 1:1,000,000. The waste collected while using the PSS contained vital bacteria, and breeding them on blood agar plates demonstrated their vitality.

In conclusion, the air-powder-water spray does not sufficiently remove the biofilm. This might explain why, in a recent review, a re-osseointegration rate of only 39%−46% was possible in animal models [21]. Considering the angle of threads, the micro-texture of the implants combined with oversized particles, and craterlike bone defects, it is more possible to follow the manufacturer´s manual (working distance of 3–5 mm and an angle of 30–60°) and to clean the microstructure properly. The electrolytic approach sterilized all the tested implants, irrespective of alloy or surface. No vital bacteria were detectable. For this reason, the null hypothesis had to be rejected. 

The electrolytic approach seems to be a promising method for treating infected dental and orthopedic implants. 

## 5. Conclusions

In this study, it was proven that electrolytic cleaning of lifelike dental implants contaminated with vital oral biofilms sterilizes the implant surfaces. One of the gold-standard methods of cleaning by air-powder-water spray failed to achieve an adequate result. Clinical testing is necessary to prove the clinical impact of these findings in dentistry and orthopedic surgery.

## Figures and Tables

**Figure 1 jcm-08-01397-f001:**
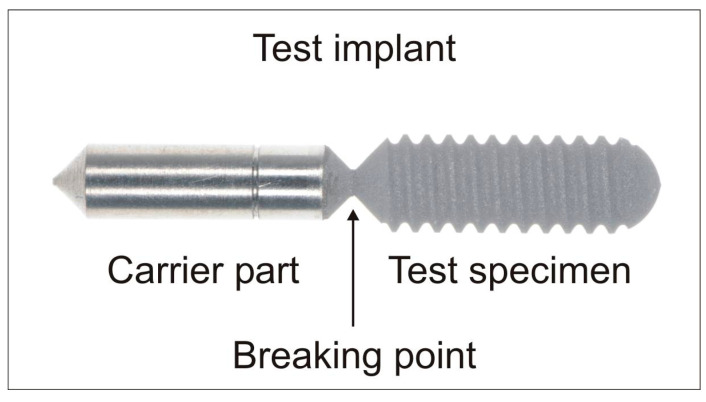
The structure of test implants.

**Figure 2 jcm-08-01397-f002:**
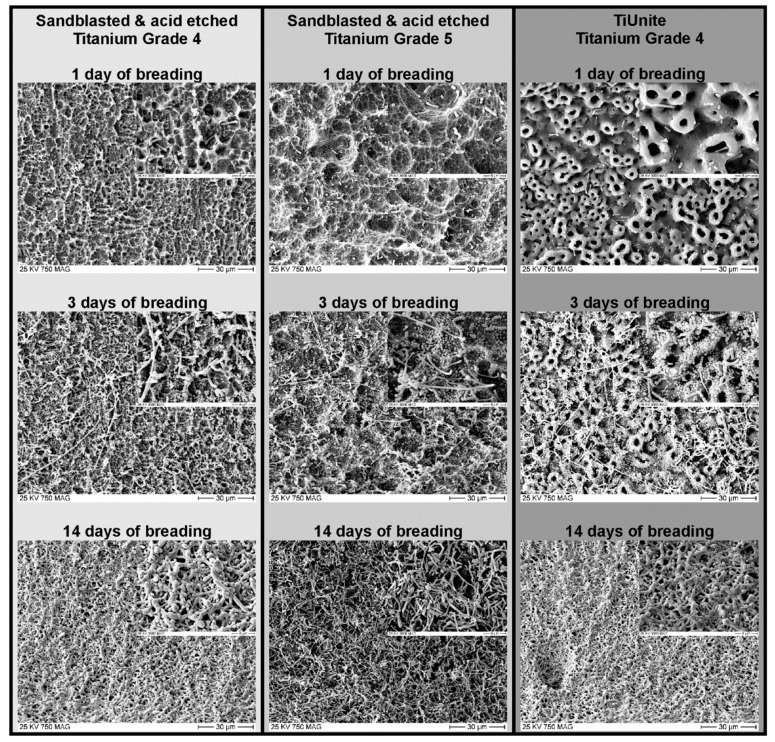
Progredient formation of biofilms on different surfaces.

**Figure 3 jcm-08-01397-f003:**
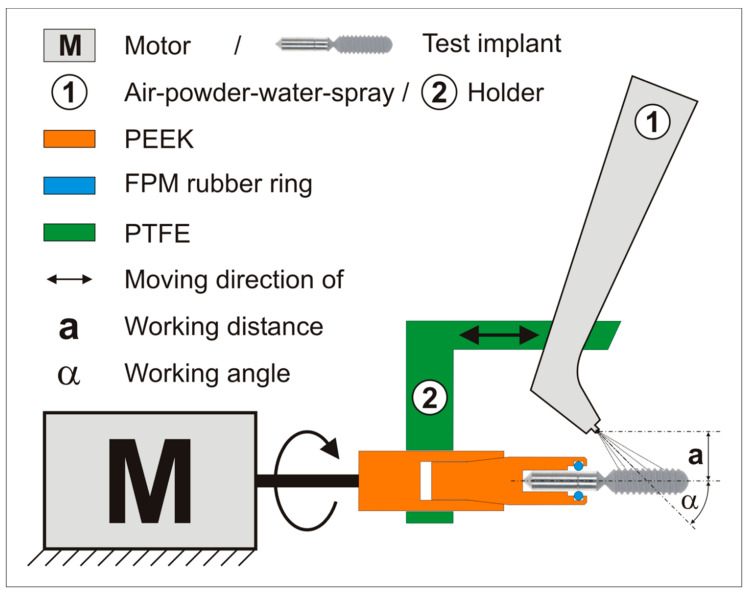
The assembly of the powder-water spray device. PEEK = polyetheretherketone, FPM = fluorine rubber, PTFE = polytetrafluorothgylene.

**Figure 4 jcm-08-01397-f004:**
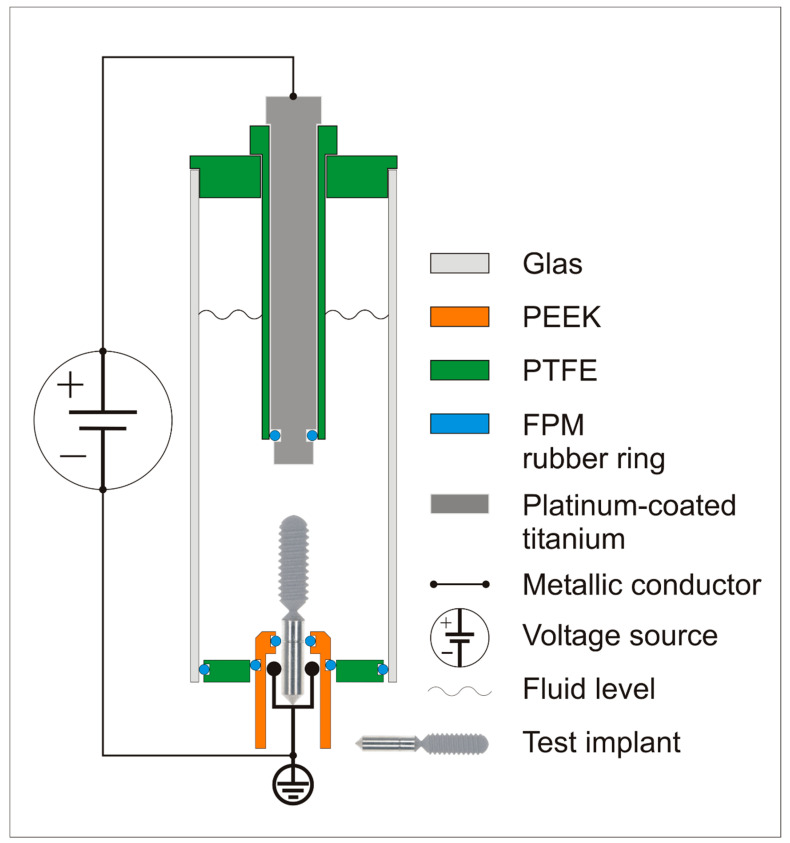
The structure of the electrolytic chamber.

**Figure 5 jcm-08-01397-f005:**
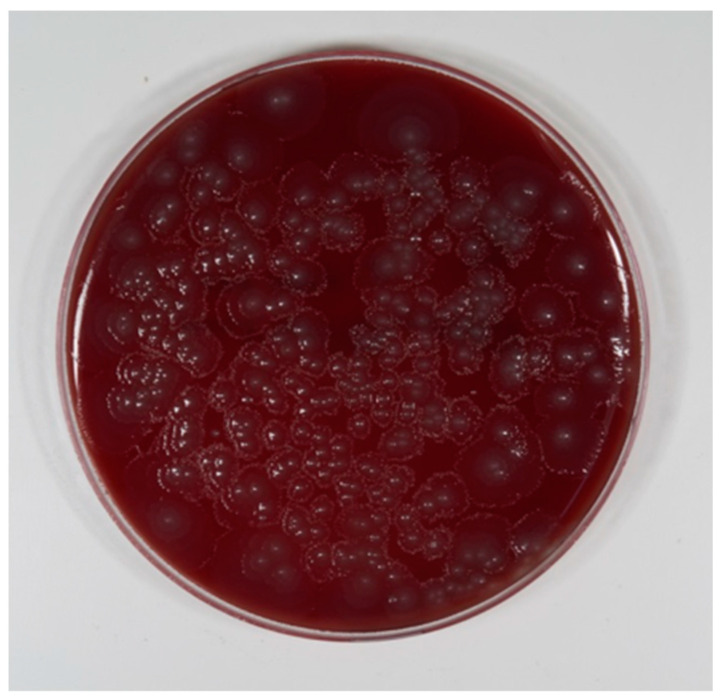
Bacterial growth with the nutritional solution of the negative control (not treated). Dilution grade 1:1,000,000.

**Figure 6 jcm-08-01397-f006:**
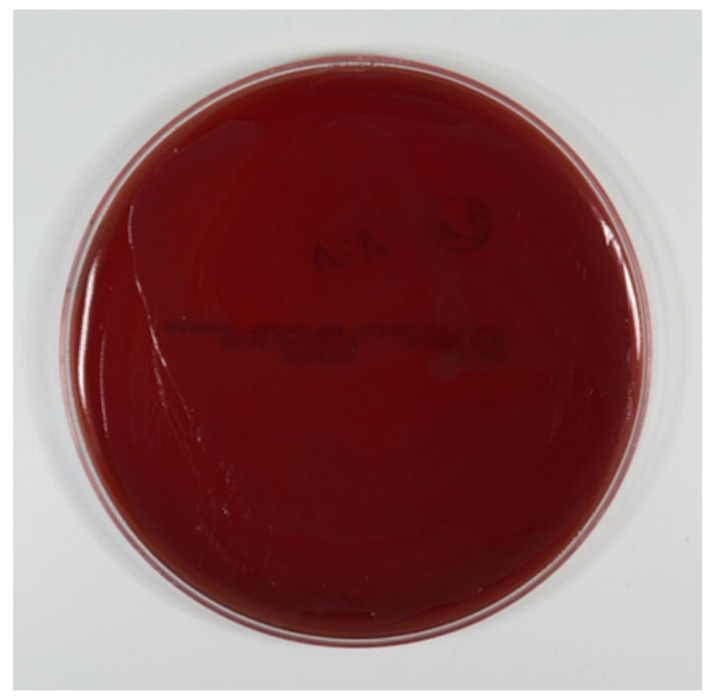
Bacterial growth after electrolytic cleaning. Dilution grade 1:1.

**Figure 7 jcm-08-01397-f007:**
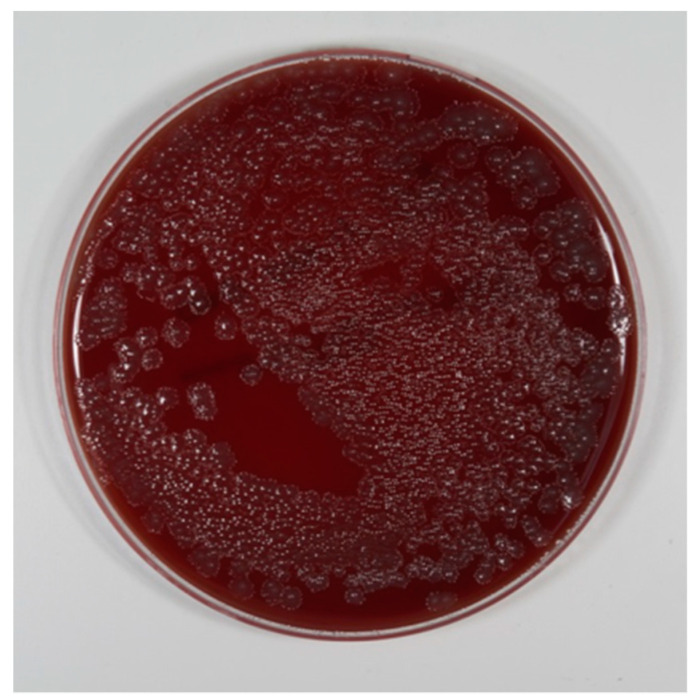
Bacterial growth after air-powder-water spray cleaning. Dilution grade 1: 1,000,000.

**Table 1 jcm-08-01397-t001:** Colony-forming unit (CFU) counts per grade of titanium, surface, dilution, and cleaning method.

Cleaning Method	Electrolytic	PPS	Electrolytic	PPS	Electrolytic	PPS
Material & Surface	Group 1: Ti Grade 4 + SAE	Group 2: Ti Grade 4 + SAE	Group 3: Ti Grade 5 + SAE	Group 4: Ti Grade 5 + SAE	Group 5: Ti Grade 4 + AO	Group 6: Ti Grade 4 + AO
Dilution	∅ CFU	∅ CFU	∅ CFU	∅ CFU	∅ CFU	∅ CFU
1:1	0	na	0	na	0	na
1:10	0	na	0	na	0	na
1:100	0	na	0	na	0	na
1:1000	0	na	0	na	0	na
1:10,000	0	na	0	na	0	na
1:100,000	0	na	0	na	0	na
1:1,000,000	0	258.1 ± 19.9	0	264.4 ±36.5	0	245.3 ± 40.7

The difference between the electrolytic approach (test groups 1, 3, and 5) and PSS (control groups 2, 4, and 6) was statistically extremely significant (*p*-value < 2.2 × 10^−16^). In control groups 2, 4, and 6 (PSS), no difference in cleaning efficacy could be detected (*p*-value = 0.3465) when comparing the different implant materials and surfaces. In test groups 1, 3, and 5 (electrolytic approach), no colony-forming units (CFUs) were detected. Therefore, there were no differences between the different implant materials and surfaces.

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
