# Peer review of "The Effect of In Vitro Electrolytic Cleaning on Biofilm-Contaminated Implant Surfaces"

_jcm, 2019, doi:10.3390/jcm8091397_

Round 1

Reviewer 1 Report

Although this in vitro study does not immediately provide clinical applicability it is a real good study with a proof of principle approach. Liked it a lot. Publication can be accepted.

In the introduction line 53 page 2, it should be mentioned that not only crater like defects but ongoing bone loss is mandatory to diagnose periimplantitis. This is also clear from the 2018 consensus report as well as the paper of Coli et al (periodontology 2000, 2017).

I suggest reference 5,6 quoted on page 3 line 58 is replaced by the coli paper because this critically commented diagnostics and prevalence.

In the results section page 7 line 198 I question whether it is possible to eject the powder correctly given the threads angels and depth. Maybe this should be commented also in the discussion as a general drawback with any method of mechanical cleaning of implants. It is impossible to reach between threads let alone to have a correct angle of the powder spraying.

Author Response

Although this in vitro study does not immediately provide clinical applicability it is a real good study with a proof of principle approach. Liked it a lot. Publication can be accepted.

In the introduction line 53 page 2, it should be mentioned that not only crater like defects but ongoing bone loss is mandatory to diagnose periimplantitis. This is also clear from the 2018 consensus report as well as the paper of Coli et al (periodontology 2000, 2017).

Comments:

We added the suggestion to the text:

Craterlike bone defects, ongoing bone loss, pus and bleeding on probing (BoP) are clinical parameters which have to be present to justify the diagnosis of PI

I suggest reference 5,6 quoted on page 3 line 58 is replaced by the coli paper because this critically commented diagnostics and prevalence.

We added the paper of Coli as required.

In the results section page 7 line 198 I question whether it is possible to eject the powder correctly given the threads angels and depth. Maybe this should be commented also in the discussion as a general drawback with any method of mechanical cleaning of implants. It is impossible to reach between threads let alone to have a correct angle of the powder spraying.

We added in Discussion:

Considering the angle of threads, the micro-texture of the implants, craterlike bone defects it is mort possible to follow manufacturer´s manual (working distance 3-5 mm and angle 30-60°) and to clean microstructure properly.

Reviewer 2 Report

Dear Author, I read with great interest your submitted manuscript, as a researcher involved in implant field. Beyond a few minor corrections, I am wandering if your proposed disinfection method may be applied (technically, biologically and safetly) to a clinical scenario. This aspect may be also discussed in your manuscript, as limitation or future prospective. A part from this, I found the results of your in vitro study very intresting and promising for future studies.

line 34-35: were counted

line 38: RC? maybe EC?

why group 5 and 6 were not explored also for grade 5 titanium?

I suggest you to cite and discuss the following paper:  

Reuse of Implant Healing Abutments: Comparative Evaluation of the Efficacy of Two Cleaning Procedures.

Stacchi C, Berton F, Porrelli D, Lombardi T.

Int J Prosthodont. 2018 Mar/Apr;31(2):161-162. doi: 10.11607/ijp.5552.

PMID: 29518810

Author Response

Dear Author, I read with great interest your submitted manuscript, as a researcher involved in implant field. Beyond a few minor corrections, I am wandering if your proposed disinfection method may be applied (technically, biologically and safetly) to a clinical scenario. This aspect may be also discussed in your manuscript, as limitation or future prospective. A part from this, I found the results of your in vitro study very intresting and promising for future studies.

We will submit a clinical paper to the journal of clinical medicine and hope it will be published at the same time.

line 34-35: were counted

changed

line 38: RC? maybe EC?

Right, changed in EC

why group 5 and 6 were not explored also for grade 5 titanium?

The TiUnite surface is only available on Titan Grade 4, unlike the sandblasted and acid etched surfaces on the market. In addition, the TiUnite surface is based on an anodic oxidation process, which is influenced by the composition of the base material (Titanium). Titanium Grade 5 contains aluminum and vanadium, which negatively affect the manufacturing process of the TiUnite surface.

I suggest you to cite and discuss the following paper:  

Reuse of Implant Healing Abutments: Comparative Evaluation of the Efficacy of Two Cleaning Procedures.

Stacchi C, Berton F, Porrelli D, Lombardi T.

Int J Prosthodont. 2018 Mar/Apr;31(2):161-162. doi: 10.11607/ijp.5552.

PMID: 29518810

Integrated
